# EntQA: Entity Linking as Question Answering

**Wenzheng Zhang, Wenyue Hua, Karl Stratos**
Department of Computer Science
Rutgers University
`{wenzheng.zhang,wenyue.hua,karl.stratos}@rutgers.edu`

## Abstract

A conventional approach to entity linking is to first find mentions in a given document and then infer their underlying entities in the knowledge base. A well-known limitation of this approach is that it requires finding mentions without knowing their entities, which is unnatural and difficult. We present a new model that does not suffer from this limitation called **EntQA**, which stands for **Ent**ity linking as **Q**uestion **A**nswering. EntQA first proposes candidate entities with a fast retrieval module, and then scrutinizes the document to find mentions of each candidate with a powerful reader module. Our approach combines progress in entity linking with that in open-domain question answering and capitalizes on pretrained models for dense entity retrieval and reading comprehension. Unlike in previous works, we do not rely on a mention-candidates dictionary or large-scale weak supervision. EntQA achieves strong results on the GERBIL benchmarking platform.

## 1 Introduction

We consider the most general form of entity linking (EL) in which a system, given a document, must both extract entity mentions and link the mentions to their corresponding entries in a knowledge base (KB). EL is a foundational building block in automatic text understanding with applications to question answering (QA) (Ferrucci, 2012), information retrieval (Xiong et al., 2017; Hasibi et al., 2016; Balog et al., 2013; Reinanda et al., 2015), and commercial recommendation systems (Yang et al., 2018; Slawski, 2015).

The output space in EL is intractably large. Any subset of all possible spans in the document linked to any KB entries (typically in the order of millions) can be a system output. To get around the intractability, existing methods decompose EL into mention detection (MD) and entity disambiguation (ED) and tackle them with varying degrees of independence. In all cases, however, the order of these two subproblems is MD followed by ED: first the system identifies potential entity mentions, and then the mentions are resolved to KB entries. Previous works either assume that mentions are given (Gupta et al., 2017), run an off-the-shelf named-entity recognition (NER) system to extract mentions and resolve them by ED (MD→ED pipeline) (Hoffart et al., 2011; Ling et al., 2015; van Hulst et al., 2020), or train an end-to-end model that jointly performs MD→ED by beam search (Kolitsas et al., 2018; De Cao et al., 2021).

A limitation of performing MD before ED is that it requires finding mentions without knowing the corresponding entities. By definition, a mention needs an entity (i.e., a mention of what?). Existing methods suffer from the dilemma of having to predict mentions before what they refer to, which is unnatural and difficult. For example, the MD→ED pipeline heuristically extracts mentions from spans of named entities found by a third-party NER system, and the performance bottleneck is often errors in MD propagated to ED. End-to-end models alleviate the problem of error propagation, but the search is only approximate and the dilemma, albeit to a lesser degree, remains.

In this work, we propose flipping the order of the two subproblems and solving ED before MD. We first find candidate entities that might be mentioned in the given document, then for each candidate find its mentions if possible. Our key observation is that while finding mentions is difficult without the knowledge of relevant entities, finding relevant entities is easy without the knowledge of their specific mentions. This simple change fundamentally solves the dilemma above since identifying mentions of a particular entity is well defined.

We cast the problem as *inverted* open-domain QA. Specifically, given a document, we use a dual encoder retriever to efficiently retrieve top-$K$ candidate entities from the KB as "questions". Then we apply a deep cross-attention reader on the document for each candidate to identify mentions of the candidate in the document as "answer spans". Unlike in standard QA, the model must predict an unknown number of questions and answers. We present a simple and effective solution based on thresholding. We call our model **EntQA**, standing for **Ent**ity linking as **Q**uestion **A**nswering.

Beyond conceptual novelty, EntQA also offers many practical advantages. First, EntQA allows us to piggyback on recent progress in dense entity retrieval and open-domain QA. For instance, we warm start EntQA with the BLINK entity retriever (Wu et al., 2020a) and ELECTRA finetuned on a QA dataset (Clark et al., 2019) to obtain an easy improvement. Second, EntQA has no dependence on a hardcoded mention-candidates dictionary which is used in previous works to reduce the search space and bias the model (Ganea & Hofmann, 2017; Kolitsas et al., 2018; De Cao et al., 2021). The dictionary is typically constructed using a large KB-specific labeled corpus (e.g., Wikipedia hyperlinks), thus having no dependence on it makes our approach more broadly applicable to KBs without such resources. Third, training EntQA is data efficient and can be done with an academic budget, in contrast with GENRE (De Cao et al., 2021) which requires industry-scale pretraining by weak supervision.

EntQA achieves strong performance on the GERBIL benchmarking platform (Röder et al., 2018). The in-domain $F_1$ score on the test portion of the AIDA-CoNLL dataset is 85.8 (2.1 absolute improvement). The macro-averaged $F_1$ score across 8 evaluation datasets is 60.5 (2.3 absolute improvement).[1] We analyze EntQA and find that its retrieval performance is extremely strong (over 98 top-100 recall on the validation set of AIDA), verifying our hypothesis that finding relevant entities without knowing their mentions is easy. We also find that the reader makes reasonable errors such as accurately predicting missing hyperlinks or linking a mention to a correct entity that is more specific than the gold label.

## 2 MODEL

Let $\mathcal{E}$ denote the set of entities in a KB associated with a text title and description. Let $\mathcal{V}$ denote the vocabulary and $\mathcal{X} = \left\{ x \in \mathcal{V}^T : 1 \le T \le T_{\max} \right\}$ the set of all documents up to length $T_{\max}$. EL is the task of mapping $x \in \mathcal{X}$ to $y \in \mathcal{P}(\mathcal{Y}(x))$ where $\mathcal{Y}(x) = \{(s, t, e) : 1 \le s \le t \le |x|, e \in \mathcal{E}\}$ is the set of all possible linked spans in $x$ and $\mathcal{P}$ is the power set. The size of the output space is $O\big(2^{T_{\max}^2 |\mathcal{E}|}\big)$ where $|\mathcal{E}|$ is typically very large (e.g., around 6 million in Wikipedia) and $T_{\max}$ can also be large (e.g., $> 3000$ in AIDA), ruling out any naive exhaustive search as a feasible approach.

EntQA decomposes EL into two subproblems: entity retrieval and question answering. More specifically, given a document $x \in \mathcal{X}$,

1. The **retriever** module retrieves top-$K$ candidate entities that might be mentioned in $x$.
2. The **reader** module extracts mentions of each candidate entity in $x$ (or rejects it), then returns a subset of globally reranked labeled mentions as the final prediction.

Our approach bears superficial similarities to a standard framework in open-domain QA that pipelines retrieval and span finding (Karpukhin et al., 2020, *inter alia*), but it has the following important differences. First, instead of retrieving passages given a question, it retrieves questions (i.e., candidate entities) given a passage. Second, even when considering a single question, there can be multiple answer spans (i.e., mentions) instead of one. Both the number of gold entities present in a document and the number of mentions of each gold entity are unknown, making this setting more challenging than standard QA in which we only need to find a single answer span for a single question on a passage.

**Input representation.** Both the retriever and the reader work with text representations of documents and entities, thus applicable to a zero-shot setting (e.g., linking to a new KB at test time by reading entity descriptions). We use the title $\phi_{\text{title}}(e) \in \mathcal{V}^+$ and the description $\phi_{\text{desc}}(e) \in \mathcal{V}^+$ to represent an entity $e \in \mathcal{E}$. Since a document $x \in \mathcal{X}$ is generally too long to encode with a Transformer encoder which has a quadratic dependency on the input length, we break it down in $m_x \in \mathbb{N}$

---

[1]Code available at: `https://github.com/WenzhengZhang/EntQA`

overlapping passages $p_1(x)\dots p_{m_x}(x)\in\mathcal{V}^L$ of length $L$ with stride $S$ (e.g., $L=32$ and $S=16$) and operate at the passage-level similarly as in QA (Alberti et al., 2019). When a document is long, individual passages may lose global information. For long documents, we find it beneficial to carry a document-level topical text $\psi_{\text{topic}}(x)\in\mathcal{V}^+$ across passages in that document (e.g., first sentence). We emphasize that we do *not* use any extra information outside the document. In our experiments we simply set $\psi_{\text{topic}}(x)=x_1\in\mathcal{V}$ (i.e., the first token in the document).

**Notation.** We write $\mathbf{enc}^\theta_S:\mathcal{V}^T\to\mathbb{R}^{d\times T}$ to denote a Transformer encoder that maps any token sequence to the same-length sequence of corresponding contextual embeddings; the symbol $S$ is used to distinguish different encoders. We assume the usual special tokens in the input popularized by BERT (Devlin et al., 2019): [CLS] to represent the whole input and [SEP] to indicate an input boundary. We write $\oplus$ to denote the text concatenation; we insert an unused token type in the vocabulary in between two texts being concatenated. We write $M_i\in\mathbb{R}^d$ to denote the $i$-th column of matrix $M\in\mathbb{R}^{d\times T}$.

## 2.1 RETRIEVER

Given a passage $p\in\mathcal{V}^+$ in document $x$ and an entity $e\in\mathcal{E}$, the retriever computes

$$P=\mathbf{enc}^\theta_P(\,\texttt{[CLS]}\,p\,\texttt{[SEP]}\,\psi_{\text{topic}}(x))$$
$$E^e=\mathbf{enc}^\theta_E(\,\texttt{[CLS]}\,\phi_{\text{title}}(e)\oplus\phi_{\text{desc}}(e)\,\texttt{[SEP]}\,)$$
$$\text{score}^\theta_{\text{retr}}(p,x,e)=P_1^\top E_1^e$$

At inference time, we precompute $E^e\in\mathbb{R}^d$ for each $e\in\mathcal{E}$ and use Faiss (Johnson et al., 2019) for fast top-$K$ retrieval.

**Training.** We train the retriever by a multi-label variant of noise contrastive estimation (NCE). Given a passage $p$ in document $x$, we have a set of multiple gold entities $\mathcal{E}(p)\subset\mathcal{E}$ that are mentioned in the passage and optimize the per-example objective

$$\max_\theta\sum_{e\in\mathcal{E}(p)}\log\left(\frac{\exp\left(\text{score}^\theta_{\text{retr}}(p,x,e)\right)}{\exp\left(\text{score}^\theta_{\text{retr}}(p,x,e)\right)+\sum_{e'\in\mathbf{N}(\mathcal{E},p)}\exp\left(\text{score}^\theta_{\text{retr}}(p,x,e')\right)}\right)\quad(1)$$

where $\mathbf{N}(\mathcal{E},p)\subset\mathcal{E}\backslash\mathcal{E}(p)$ is a set of negative examples that excludes *all* gold entities $\mathcal{E}(p)$. The objective effectively constructs $|\mathcal{E}(p)|$ independent NCE instances, each of which treats a gold entity as the only correct answer while ensuring that other gold entities are not included in negative examples. We obtain 90% of $\mathbf{N}(\mathcal{E},p)$ by sampling entities uniformly at random from $\mathcal{E}\backslash\mathcal{E}(p)$ and 10% by hard negative mining (i.e., using highest-scoring incorrect entities under the model), which is well known to be beneficial in entity retrieval (Gillick et al., 2019; Wu et al., 2020a; Zhang & Stratos, 2021).

## 2.2 READER

Let $e_{1:K}=(e_1\dots e_K)\in\mathcal{E}^K$ denote $K$ candidate entities for a passage $p$ in document $x$. For each $k\in\{1\dots K\}$, the reader computes a joint encoding of $(p,x,e_k)$ by

$$H^k=\mathbf{enc}^\theta_H(\,\texttt{[CLS]}\,p\oplus\psi_{\text{topic}}(x)\,\texttt{[SEP]}\,\phi_{\text{title}}(e_k)\oplus\phi_{\text{desc}}(e_k)\,\texttt{[SEP]}\,)$$

then defines a conditional distribution over mention spans of $e_k$ in $p$ by

$$p^\theta_{\text{start}}(s|p,x,e_k)=\frac{\exp\left(w_{\text{start}}^\top H_s^k\right)}{\sum_{i=1}^{|p|+1}\exp\left(w_{\text{start}}^\top H_i^k\right)}\qquad\forall s\in\{1\dots|p|+1\}$$

$$p^\theta_{\text{end}}(t|p,x,e_k)=\frac{\exp\left(w_{\text{end}}^\top H_t^k\right)}{\sum_{i=1}^{|p|+1}\exp\left(w_{\text{end}}^\top H_i^k\right)}\qquad\forall t\in\{1\dots|p|+1\}$$

$$p^\theta_{\text{span}}(s,t|p,x,e_k)=p^\theta_{\text{start}}(s|p,x,e_k)\times p^\theta_{\text{end}}(t|p,x,e_k)\qquad\forall s,t\in\{1\dots|p|+1\}$$

where $w_{\text{start}}, w_{\text{end}} \in \mathbb{R}^d$ are additional parameters. The reader also multitasks reranking: it uses $w_{\text{rerank}} \in \mathbb{R}^d$ to define a conditional distribution over candidate entities by

$$p_{\text{rerank}}^{\theta}(e_k|p, x, e_{1:K}) = \frac{\exp\left(w_{\text{rerank}}^{\top} H_1^k\right)}{\sum_{k'=1}^{K} \exp\left(w_{\text{rerank}}^{\top} H_1^{k'}\right)} \qquad \forall k \in \{1 \ldots K\}$$

**Training.** We obtain candidates $e_{1:K}$ from a fully trained retrieval module to make training consistent with test time. During training, we always include all gold entities as candidates (i.e., $\mathcal{E}(p) \subset e_{1:K}$). Let $\mathcal{M}(p, e)$ denote the set of gold mention spans of $e \in \mathcal{E}$ in $p$; if $e$ is not present in $p$, we define $\mathcal{M}(p, e) = \{(1, 1)\}$. We optimize the per-example objective

$$\max_{\theta} \sum_{k=1}^{K} \mathbb{1}(e_k \in \mathcal{E}(p)) \log p_{\text{rerank}}^{\theta}(e_k|p, x, e_{1:K}) + \sum_{(s,t) \in \mathcal{M}(p, e_k)} \log p_{\text{span}}^{\theta}(s, t|p, x, e_k) \quad (2)$$

where $\mathbb{1}(A)$ is the indicator function equal to one if $A$ is true and zero otherwise. Note that the reader is trained to predict the [CLS] span for incorrect entities.

## 2.3 INFERENCE

At test time, we process a new document $x \in \mathcal{X}$ in passages $p \in \mathcal{V}^L$ independently as follows:

1. Retrieve top-$K$ highest scoring entities $e_{1:K}$ under $\text{score}_{\text{retr}}^{\theta}(p, x, e)$.
2. For each candidate $k$, extract top-$P$ most likely mention spans $(s_1^k, t_1^k) \ldots (s_P^k, t_P^k)$ under $p_{\text{span}}^{\theta}(s, t|p, x, e_k)$ while discarding any span less probable than $(1, 1)$.
3. Return a subset of the surviving labeled mentions $(s, t, e_k)$ with $p_{\text{rerank}}^{\theta}(e_k|p, x, e_{1:K}) \times p_{\text{span}}^{\theta}(s, t|p, x, e_k) > \gamma$ as the final prediction.

We do not apply any further processing to combine passage-level predictions other than merging duplicate labeled spans $(s, t, e)$ in the overlapping sections. This inference scheme is simple yet effective. For each candidate entity, the reader scrutinizes the passage with deep cross-attention to see if there are any mentions of the entity and has a chance to reject it by predicting $(1, 1)$. The reader delays its final decision until it has processed all candidates to globally reconsider labeled mentions with ranking probabilities. Figure 1 shows a successful prediction on a passage from the validation portion of AIDA.

## 3 EXPERIMENTS

We evaluate EntQA on the GERBIL benchmarking platform (Röder et al., 2018), which offers reliable comparison with state-of-the-art EL methods on numerous public datasets.

## 3.1 SETTING

**Datasets.** We follow the established practice and report the InKB Micro $F_1$ score on the in-domain and out-of-domain datasets used in De Cao et al. (2021). Specifically, we use the AIDA-CoNLL dataset (Hoffart et al., 2011) as the in-domain dataset: we train EntQA on the training portion of AIDA, use the validation portion (AIDA-A) for development, and reserve the test portion (AIDA-B) for in-domain test performance. We use seven out-of-domain test sets: MSNBC, Derczynski (Der) (Derczynski et al., 2015), KORE 50 (K50) (Hoffart et al., 2012), N3-Reuters-128 (R128), N3-RSS-500 (R500) (Röder et al., 2014), and OKE challenge 2015 and 2016 (OKE15 and OKE16) (Nuzzolese et al., 2015). We refer to Table 6 in Kolitsas et al. (2018) for the datasets' statistics. For the KB, we use the 2019 Wikipedia dump provided in the KILT benchmark (Petroni et al., 2021), which contains 5.9 million entities.

**Model details.** We initialize the passage encoder $\textbf{enc}_P^{\theta}$ and the entity encoder $\textbf{enc}_E^{\theta}$ in the retriever module with independent BLINK retrievers pretrained on Wikipedia hyperlinks (Wu et al., 2020a) and optimize the NCE objective (1) with hard negative mining. We initialize the joint encoder

**Passage**

After bowling [Somerset]$_3$ out for 83 on the opening morning at [**Grace Road**]$_2$, [**Leicestershire**]$_1$ extended their first innings by 94 runs before being bowled out for 296 with [**England**]$_{11}$

**Top-$K$ candidate entities**

   1. **Leicestershire County Cricket Club**
   2. **Grace Road**
   3. **Somerset County Cricket Club**
✗  4. Durham County Cricket Club
✗  5. Nottinghamshire County Cricket Club
✗  6. Derbyshire County Cricket Club
✗  7. Warwickshire County Cricket Club
✗  8. Leicestershire
✗  9. Worcestershire County Cricket Club
✗ 10. Yorkshire County Cricket Club
  11. **England cricket team**
✗ 12. Marylebone Cricket Club
✗ 13. Sussex County Cricket Club
✗ 14. Kent County Cricket Club
✗ 15. Leicester
✗ 16. Aylestone Road
✗ 17. County Cricket Ground, Derby
   ⋮

Figure 1: Example prediction by EntQA taken from AIDA-A. Given a passage, the retriever module ranks $K$ candidate entities, then the reader module finds mentions of each entity or rejects it (marked by ✗). Both modules use entity descriptions (not shown). In this example, it predicts the span "England" for the 11th candidate `England cricket team` but rejects the 35th candidate `England` (the country).

$\mathbf{enc}_H^\theta$ in the reader module with ELECTRA-large (Clark et al., 2019) finetuned on SQuAD 2.0 (Rajpurkar et al., 2018) and optimize the reader objective (2). We break up each document $x \in \mathcal{X}$ into overlapping passages of length $L = 32$ with stride $S = 16$ under WordPiece tokenization. For each passage in $x$, we concatenate the input with the first token of the document $\psi_{\text{topic}}(x) = x_1$, which corresponds to the topic in AIDA but not in other datasets. We use 64 candidate entities in training for both the retriever and the reader; we use 100 candidates at test time. We predict up to $P = 3$ mention spans for each candidate entity. We use $\gamma = 0.05$ as the threshold in all experiments, chosen after trying values 0.01, 0.1, and 0.05 on the validation set. Additional experiments on automatically tuning $\gamma$ are discussed in Appendix A. For optimization, we use Adam (Kingma & Ba, 2015) with learning rate 2e-6 for the retriever and 1e-5 for the reader; we use a linear learning rate decay schedule with warmup proportion 0.06 over 4 epochs for both modules. The batch size is 4 for the retriever and 2 for the reader. The retriever is trained on 4 GPUs (A100) for 9 hours; the reader is trained on 2 GPUs for 6 hours.

**Baselines.** We compare with state-of-the-art EL systems that represent a diverse array of approaches. Hoffart et al. (2011) and van Hulst et al. (2020) use the MD→ED pipeline; despite the limitation of pipelining MD with ED, the latter achieve excellent performance by solving MD with a strong NER system (Akbik et al., 2018). Kolitsas et al. (2018) use an end-to-end model that sequentially performs MD and ED; to make the problem tractable, they drastically prune the search space with a mention-candidates dictionary and the model score. De Cao et al. (2021) propose GENRE, a sequence-to-sequence model for EL. The model conditions on the given document and autoregressively generates a labeled version of the document by at each position either copying a token, starting or ending a mention span, or, if the previous generation was the end of a mention $m$, generating the entity title associated with $m$ token by token. At inference time, GENRE critically relies on a prefix tree (aka. trie) derived from Wikipedia to constrain the beam search so that it produces a valid entity title in the KB. Since each beam element must first predict a mention before predicting an entity, unless the beam size is unbounded so that every labeled span is considered, GENRE will suffer from MD errors propagating to ED.

## 3.2 RESULTS

Table 1 shows the main results. EntQA achieves the best in-domain test $F_1$ score for AIDA (+2.1) and is also performant on out-of-domain datasets (+3.8 on KORE 50 and +7.4 on N3-Reuters-128, close second-best on Derczynski and N3-RSS-500). The performance is lower on OKE15 and OKE16 for the same reason pointed out by De Cao et al. (2021): these datasets are annotated with

Table 1: InKB Micro $F_1$ on the in-domain and out-of-domain test sets on the GERBIL benchmarking platform. For each dataset, **bold** indicates the best model and underline indicates the second best.

| | In-domain | | Out-of-domain | | | | | | |
| Method | AIDA | MSNBC | Der | K50 | R128 | R500 | OKE15 | OKE16 | Avg |
|---|---|---|---|---|---|---|---|---|---|
| Hoffart et al. (2011) | 72.8 | 65.1 | 32.6 | 55.4 | 46.4 | **42.4** | **63.1** | 0.0 | 47.2 |
| Steinmetz & Sack (2013) | 42.3 | 30.9 | 26.5 | 46.8 | 18.1 | 20.5 | 46.2 | 46.4 | 34.7 |
| Moro et al. (2014) | 48.5 | 39.7 | 29.8 | 55.9 | 23.0 | 29.1 | 41.9 | 37.7 | 38.2 |
| Kolitsas et al. (2018) | 82.4 | 72.4 | 34.1 | 35.2 | 50.3 | 38.2 | 61.9 | 52.7 | 53.4 |
| Broscheit (2019) | 79.3 | - | - | - | - | - | - | - | |
| Martins et al. (2019) | 81.9 | - | - | - | - | - | - | - | |
| van Hulst et al. (2020) | 80.5 | 72.4 | 41.1 | 50.7 | 49.9 | 35.0 | **63.1** | **58.3** | 56.4 |
| De Cao et al. (2021) | 83.7 | **73.7** | **54.1** | 60.7 | 46.7 | 40.3 | 56.1 | 50.0 | 58.2 |
| **EntQA** | **85.8** | 72.1 | 52.9 | **64.5** | **54.1** | 41.9 | 61.1 | 51.3 | **60.5** |

coreference (i.e., they contain pronouns and common nouns linked to entities) which our model is not trained for, while many other systems have a component in their pipelines to handle these cases. We hypothesize that the performance on MSNBC is lagging because it has long documents (544 words per document on average) which are processed in relatively short passages under EntQA due to our computational constraints. Overall, EntQA achieves the best macro-averaged $F_1$ score across the 8 evaluation datasets (+2.3).

The inference runtime of EntQA is clearly linear in the number of candidate entities $K$. To get a sense of speed, we compared the runtime of EntQA with that of GENRE on the AIDA validation set using 1 GPU on the same machine. GENRE took 1 hour and 10 minutes, excluding 31 minutes to first build a prefix tree. EntQA took 20 minutes with $K = 100$, 10 minutes with $K = 50$, and 4 minutes with $K = 20$, excluding 1 hour to first index entity embeddings, yielding $F_1$ scores 87.3, 87.4, and 87.0. Interestingly, we can obtain a significant speedup at a minor cost in performance by decreasing $K$. We believe this can be a useful feature of the model in controlling the speed-performance tradeoff.

We note that there is an issue of using different editions of Wikipedia between the systems. For instance, Hoffart et al. (2011) use the 2010 dump, van Hulst et al. (2020) and we use the 2019 dump, whereas Kolitsas et al. (2018) and De Cao et al. (2021) use the 2014 dump (even though the latter use the 2019 dump for pretraining). Thus there is a concern that differences in performance are due to different snapshots of Wikipedia. While we consider it out of scope in our work to fully address this concern, we find that using different editions of Wikipedia does not fundamentally change the performance of EntQA, which is consistent with GERBIL's intent of being KB-agnostic. For instance, we obtained the same validation $F_1$ on AIDA with our model trained on either the 2014 or 2019 dump. We use the KILT edition of Wikipedia mainly for convenience.

### 3.2.1 OTHER PRACTICAL HIGHLIGHTS

**No dictionary.** EntQA has no dependence on a mention-candidates dictionary. All previous works rely on a dictionary $\mathcal{D} : \mathcal{V}^+ \to \mathcal{P}(\mathcal{E})$ that maps a mention string $m$ to a small set of candidate entities $e \in \mathcal{E}$ associated with empirical conditional probabilities $\hat{p}_{e|m} > 0$ (Hoffart et al., 2011, *inter alia*). For instance, it is an essential component of the search procedure in the end-to-end model of Kolitsas et al. (2018). While not mentioned in the paper or on the GitHub repository, GENRE (De Cao et al., 2021) also uses the dictionary from Kolitsas et al. (2018) in their prefix tree to constrain the beam search (personal communication with one of the authors of the paper). Constructing such a dictionary typically assumes the existence of a large KB-specific labeled corpus (e.g., internal links in Wikipedia). EntQA is thus more broadly applicable to KBs without such resources (e.g., for small domain-specific KBs).

**No model-specific pretraining.** EntQA does not require model-specific pretraining; it only uses standard pretrained Transformers for initialization and is directly finetuned on AIDA. This is in contrast with GENRE which requires industry-scale pretraining by weak supervision. Specifically,

Table 2: Ablation study for the retriever module. Each line makes a single change from the baseline retriever used in Table 1. We also compare with BM25.

| Retriever | Val R@100 |
|---|---|
| Baseline | 98.2 |
| – Omit excluding other gold entities in the normalization term of NCE | 82.7 |
| – Train by optimizing the marginal log-likelihood | 83.8 |
| – Initialize with BERT-large | 94.4 |
| – Omit hard negatives in NCE (i.e., negative examples are all random) | 94.4 |
| – Omit the document-level information $x_1$ in the passage representation | 96.6 |
| BM25 | 36.6 |

Table 3: Ablation study for the reader module. Each line makes a single change from the baseline reader used in Table 1. Candidate entities are obtained from the baseline retriever in Table 2 (except the oracle experiment).

| Reader | Val $F_1$ |
|---|---|
| Baseline | 87.3 |
| – Initialize with BERT-large | 85.6 |
| – Train by optimizing the marginal log-likelihood | 86.9 |
| – Initialize with ELECTRA-large (not finetuned on SQuAD 2.0) | 88.4 |
| – Omit the reranking probabilities $p_{\text{rerank}}^{\theta}$ (i.e., only use span probabilities) | 87.9 |
| – Omit the document-level information $x_1$ in the input passage representation | 87.5 |
| Oracle experiment: use gold entities as the only candidate entities | 94.9 |

GENRE is trained by finetuning BART (Lewis et al., 2020) on autoregressive EL training examples constructed from all Wikipedia abstract sections on 64 GPUs for 30 hours, followed by finetuning on AIDA. Thus training GENRE from scratch is beyond the means of most academic researchers, making it difficult to make substantial changes to the model. EntQA can be trained with academic resources and outperforms GENRE.

## 3.3 ABLATION STUDIES

The final form of EntQA in Section 3.2 is the result of empirically exploring various modeling and optimization choices during development. We present an ablation study to illustrate the impact of these choices.

**Retriever**    Table 2 shows an ablation study for the retriever module. We report top-100 recall (R@100) on the validation set of AIDA. The baseline retriever is initialized with BLINK (Wu et al., 2020a), uses the passage representation `[CLS]`$p$`[SEP]`$x_1$, and is trained by optimizing the multi-label variant of NCE (1) that considers one gold entity at a time by excluding others in the normalization term. We see that the baseline retriever has an extremely high recall (98.2), confirming our hypothesis that it is possible to accurately infer relevant entities in a passage without knowing where they are mentioned. We also see that it is very important to use the proposed multi-label variant of NCE instead of naive NCE that normalizes over all gold entities, which results in a massive decrease in recall (82.7). We consider optimizing the marginal log-likelihood (i.e., the log of the sum of the probabilities of gold entities, rather than the sum of the log), but it yields much worse performance (83.8). It is helpful to initialize with BLINK rather than BERT-large, use hard negatives in NCE, and append $x_1$ to input passages. Table 2 additionally shows the BM25 recall, which is quite poor (36.6). Upon inspection, we find that BM25 fails to retrieve diverse entities. For instance, a passage on cricket may have diverse gold entities such as an organization (`Leicestershire County Cricket Club`), location (`London`), and person (`Phil Simmons`), but the top entities under BM25 are dominated by person entities (`Alan Shipman`, `Dominique Lewis`, etc.). This shows the necessity of explicitly training a retriever to prioritize diversity in our problem.

Table 4: Categorizing errors on the validation set passages. The number of passages in each category is given in parentheses. **G** refers to the gold annotation; **P** refers to the predicted annotation.

| Error | Examples (text snippets) |
|---|---|
| Over (443) | **G**: england fast bowler [martin mccague]$_{\text{Martin McCague}}$ (Fill in missing mentions) |
| | **P**: [england]$_{\text{England cricket team}}$ fast bowler [martin mccague]$_{\text{Martin McCague}}$ |
| | **G**: duran, 45, takes on little - known [mexican]$_{\text{Mexico}}$ |
| | **P**: [duran]$_{\text{Roberto Durán}}$, 45, takes on little - known [mexican]$_{\text{Mexico}}$ |
| Under (474) | **G**: second innings before [simmons]$_{\text{Phil Simmons}}$ stepped in (Bad threshold) |
| | **P**: second innings before simmons stepped in |
| | **G**: [ato boldon]$_{\text{Ato Boldon}}$ - lpr - [trinidad]$_{\text{Trinidad}}$ - rpr - 20. |
| | **P**: [ato boldon]$_{\text{Ato Boldon}}$ - lpr - trinidad - rpr - 20. |
| Neither (378) | **G**: match against yorkshire at [headingley]$_{\text{Headingly}}$ (Ambiguous entity) |
| | **P**: match against yorkshire at [headingley]$_{\text{Headingly Stadium}}$ |
| | **G**: at the [oval]$_{\text{The Oval}}$, surrey captain chris lewis (Ambiguous span) |
| | **P**: at [the oval]$_{\text{The Oval}}$, surrey captain chris lewis |
| | **G**: scores in [english]$_{\text{England}}$ county championship matches (Others) |
| | **P**: scores in [english county championship]$_{\text{County Championship}}$ matches |

**Reader** Table 3 shows an ablation study for the reader module. We report $F_1$ on the validation set of AIDA. The baseline reader is initialized with ELECTRA-large (Clark et al., 2019) finetuned on SQuAD 2.0, uses the joint passage-entity input representation $[\text{CLS}] p \oplus x_1 [\text{SEP}] \phi_{\text{title}}(e) \oplus \phi_{\text{desc}}(e) [\text{SEP}]$, and is trained by optimizing (2). Candidate entities are obtained from the baseline retriever in Table 2. We see that BERT is less performant than ELECTRA for reader initialization, consistent with findings in the QA literature (Yamada et al., 2021). Training by optimizing the marginal log-likelihood is comparable to (2). Interestingly, we find that we can fit the reader just as well without using a SQuAD-finetuned ELECTRA, ranking probabilities, or $x_1$ in passages. However, in our preliminary investigation we found that these variants generalized slightly worse outside the training domain, thus we kept our original choice. We discuss other choices of document-level information in Appendix B. Lastly, we conduct an oracle experiment in which we provide only gold entities as candidates to the reader. In this scenario, the reader is very accurate (94.9 $F_1$), suggesting that the main performance bottleneck is correctly distinguishing gold vs non-gold entities from the candidates. We investigate this issue more in depth in the next section.

## 3.4 ERROR ANALYSIS

To better understand the source of errors made by EntQA, we examine passages in the validation set for which the model's prediction is not completely correct. We partition them into three types: (1) over-predicting (i.e., the gold mentions are a strict subset of the predicted mentions), (2) under-predicting (i.e., the predicted mentions are a strict subset of the gold mentions), and (3) neither over- nor under-predicting. Table 4 shows examples of each error type. We find that over-predicting often happens because the model correctly "fills in" entity mentions missing in the gold annotation. Under-predicting happens most likely because the threshold value is too large to catch certain mentions. Finally, many errors that are neither over- nor under-predicting are largely due to annotation noise. For instance, the predicted entity `Headingly Stadium` is a correct and more specific entity for the span "headingley" than the gold entity `Headingly` (a suburb); the predicted span "the oval" is more suitable, or at least as correct as, the gold span "oval" for the entity `The Oval`.

We also consider distinguishing MD errors from ED errors on the validation set. EntQA obtains 87.5 overall $F_1$. When we only measure the correctness of mention spans (equivalent to treating all entity predictions as correct), we obtain 92.3 $F_1$. When we only measure the correctness of rejecting or accepting candidate entities, we obtain 64.5 $F_1$ at the passage level and 89.3 $F_1$ at the document level (i.e., consider the set of candidates from all passages). The reader's relatively low passage-level $F_1$ in rejecting or accepting candidates is consistent with the the oracle experiment in Table 3. That is, the main performance bottleneck of EntQA is discriminating gold vs non-gold entities from the candidates, though this should be taken with a grain of salt given the noise in annotation illustrated in Table 4.

## 4 RELATED WORK

Our work follows the recent trend of formulating language tasks as QA problems, but to our knowledge we are the first to propose reduction to inverted open-domain QA. Most previous works supply questions as input to the system, along with passages in which answer spans are found. They differ only in question formulation, for instance a predicate in semantic role labeling (He et al., 2015), a relation type along with its first argument in KB completion (Levy et al., 2017; Li et al., 2019), an entity category in (nested) NER (Li et al., 2020), an auxiliary verb or a *wh*-expression in ellipsis resolution (Aralikatte et al., 2021), and other task-specific questions (McCann et al., 2018). In contrast, we solve question formulation as part of the problem by exploiting recent advances in dense text retrieval.

A notable exception is CorefQA (Wu et al., 2020b), from which we take direct inspiration. In this approach, the authors formulate coreference resolution as QA in which questions are coreferring spans and answers are the spans' antecedents (i.e., earlier spans that belong to the same coreference cluster). Since coreferring spans are unknown, the authors rely on the end-to-end coreference resolution model of Lee et al. (2017) that produces candidate spans by beam search. In contrast, EntQA handles varying numbers of questions in a simpler framework of text retrieval.

As in this work, some previous works propose methods to handle varying numbers of answer spans for a given question. But their methods are based on one-vs-all classification (i.e., each label is associated with a token-level binary classifier) or reduction to tagging (i.e., spans are expressed as a BIO-label sequence) (Wu et al., 2020b; Li et al., 2019; 2020). We found these methods to be ineffective in preliminary experiments, and instead develop a more effective inference scheme in which the model delays its final prediction to the end for global reranking (Section 2.3).

We discuss pros and cons of EntQA vs other models in practice. While EntQA outperforms GENRE without large-scale weakly supervised pretraining, it involves dense retrieval which incurs a large memory footprint to store and index dense embeddings as pointed out by De Cao et al. (2021). But it can be done on a single machine with ample RAM (ours has 252G) which is cheap. Bypassing dense retrieval is a unique strength of the autoregressive approach of GENRE and orthogonal to ours; we leave combining their strengths as future work. Our model requires a threshold $\gamma$ for inference, but we find that it is easy to pick a good threshold; we also argue that it can be a useful feature in a real-world setting in which the practitioner often needs a customized trade-off between precision and recall. The threshold-based inference implies another unique feature of EntQA not explored in this work: it can naturally handle nested entity mentions. We leave nested linking as future work.

## 5 CONCLUSIONS

Existing methods for entity linking suffer from the dilemma of having to predict mentions without knowing the corresponding entities. We have presented EntQA, a new model that solves this dilemma by predicting entities first and then finding their mentions. Our approach is based on a novel reduction to inverse open-domain QA in which we retrieve an unknown number of questions (candidate entities) and predict potentially multiple answer spans (mentions) for each question. Our solution is a simple pipeline that takes full advantage of progress in text retrieval and reading comprehension. EntQA achieves new state-of-the-art results on the GERBIL benchmarking platform without relying on a KB-specific mention-candidates dictionary or expensive model-specific pretraining.

ACKNOWLEDGMENTS

This work was supported by the Google Faculty Research Awards Program.

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

Table 5: GERBIL test scores with and without using the first document token as document-level topical information.

| Topic Info | AIDA | MSNBC | Der | K50 | R128 | R500 | OKE15 | OKE16 | Avg |
|---|---|---|---|---|---|---|---|---|---|
| None | 81.7 | **72.2** | 52.5 | 64.2 | 54.0 | 40.3 | 59.0 | 48.9 | 59.1 |
| First token | **85.8** | 72.1 | **52.9** | **64.5** | **54.1** | **41.9** | **61.1** | **51.3** | **60.5** |

Pranav Rajpurkar, Robin Jia, and Percy Liang. Know what you dont know: Unanswerable questions for squad. In *Proceedings of the 56th Annual Meeting of the Association for Computational Linguistics (Volume 2: Short Papers)*, pp. 784–789, 2018.

Ridho Reinanda, Edgar Meij, and Maarten de Rijke. Mining, ranking and recommending entity aspects. In *Proceedings of the 38th International ACM SIGIR Conference on Research and Development in Information Retrieval*, pp. 263–272, 2015.

Michael Röder, Ricardo Usbeck, Sebastian Hellmann, Daniel Gerber, and Andreas Both. $N^3$-a collection of datasets for named entity recognition and disambiguation in the nlp interchange format. In *LREC*, pp. 3529–3533, 2014.

Michael Röder, Ricardo Usbeck, and Axel-Cyrille Ngonga Ngomo. Gerbil–benchmarking named entity recognition and linking consistently. *Semantic Web*, 9(5):605–625, 2018.

Bill Slawski. How google uses named entity disambiguation for entities with the same names, September 2015. URL https://www.seobythesea.com/2015/09/disambiguate-entities-in-queries-and-pages/. Accessed: 2021-09-27.

Nadine Steinmetz and Harald Sack. Semantic multimedia information retrieval based on contextual descriptions. In *Extended Semantic Web Conference*, pp. 382–396. Springer, 2013.

Johannes M van Hulst, Faegheh Hasibi, Koen Dercksen, Krisztian Balog, and Arjen P de Vries. Rel: An entity linker standing on the shoulders of giants. In *Proceedings of the 43rd International ACM SIGIR Conference on Research and Development in Information Retrieval*, pp. 2197–2200, 2020.

Ledell Wu, Fabio Petroni, Martin Josifoski, Sebastian Riedel, and Luke Zettlemoyer. Scalable zero-shot entity linking with dense entity retrieval. In *Proceedings of the 2020 Conference on Empirical Methods in Natural Language Processing (EMNLP)*, pp. 6397–6407, 2020a.

Wei Wu, Fei Wang, Arianna Yuan, Fei Wu, and Jiwei Li. Corefqa: Coreference resolution as query-based span prediction. In *Proceedings of the 58th Annual Meeting of the Association for Computational Linguistics*, pp. 6953–6963, 2020b.

Chenyan Xiong, Jamie Callan, and Tie-Yan Liu. Word-entity duet representations for document ranking. In *Proceedings of the 40th International ACM SIGIR conference on research and development in information retrieval*, pp. 763–772, 2017.

Ikuya Yamada, Akari Asai, and Hannaneh Hajishirzi. Efficient passage retrieval with hashing for open-domain question answering. *arXiv preprint arXiv:2106.00882*, 2021.

Yi Yang, Ozan İrsoy, and Kazi Shefaet Rahman. Collective entity disambiguation with structured gradient tree boosting. In *Proceedings of the 2018 Conference of the North American Chapter of the Association for Computational Linguistics: Human Language Technologies, Volume 1 (Long Papers)*, pp. 777–786, 2018.

Wenzheng Zhang and Karl Stratos. Understanding hard negatives in noise contrastive estimation. In *Proceedings of the 2021 Conference of the North American Chapter of the Association for Computational Linguistics: Human Language Technologies*, pp. 1090–1101, 2021.

## A    THRESHOLD OPTIMIZATION

To see if it is possible to improve over the static threshold value $\gamma = 0.05$, we tried automatically calibrating $\gamma$ based on the AIDA validation performance by considering every effective threshold obtained from a sorted list of probabilities of labeled mentions. The best threshold was $\gamma = 0.03146$. The validation F1 score improved from 87.32 to 87.75, and the GERBIL test score improved from 60.46 to 60.55. Thus threshold optimization can yield a minor improvement, but overall we find that EntQA is robust to choices of threshold in a reasonable range.

## B    DOCUMENT-LEVEL INFORMATION

We explored various ways of injecting document-level information in paragraphs. We tried the first token, the first sentence, and a continuous topic embedding (obtained by averaging all token embeddings in the document). We settled on the first-token version because it gave the best performance. For many of the GERBIL datasets, however, we obtain almost the same performance with or without the topic information. As it is somewhat dataset-specific (e.g., the first word in AIDA is always the topic word), we leave it as an option in our model for the user to decide. Table 5 shows the GERBIL performance without any topic information vs with the first token in the document.

