# OpenReview forum: "EntQA: Entity Linking as Question Answering"
_ICLR.cc/2022/Conference — ICLR 2022 Spotlight_

### Official Review · Reviewer_Uses · 2021-10-22

**Correctness:** 3
**Technical Novelty And Significance:** 3
**Empirical Novelty And Significance:** 2
**Recommendation:** 8
**Confidence:** 4

**Main Review:**

This paper is clearly written, has a working original idea, and provides a great new option for entity linking that has empirical promise. It's in good shape, and I don't have a lot of criticisms. Just a comments and questions below:

Page 1 - "End-to-end models alleviate the problem of error propagation, but the search is only approximate and the dilemma, albeit to a lesser degree, remains" - I think this sentence need some additional justification. Sure, search is approximate. That on its own doesn't mean that the dilemma remains. How would you establish that the dilemma remains?

Page 3 - "In our experiments we simply set ψtopic(x) = x1 ∈V (i.e., the first token in the document)." That seems odd. The first token may not be very useful. It could be "The". Could you use the encoded first phrase?

Page 3 - "We write ⊕to denote the text concatenation (we insert a special symbol to represent the concatenation)" I don't understand the parenthetical. Can you elaborate?

The bad threshold suggest that the model is perhaps not calibrated in some key way. Could an explicit calibration step on the validation set help fix these errors? More broadly, the fixed threshold seems like the weakest part of the system. What ideas do you have about making it more flexible or adaptable? Did you try other approaches to thresholding and found them to not work? These findings would be good to include in the paper.

**Summary Of The Paper:**

Standard entity linking first does mention detection and then the actual linking of mentions to entities in the knowledge base, and the authors propose to flip this by using open domain QA. They first use the text to find candidate entities in the knowledge base. Then for each of these candidates, they look for possible mentions in the text. The model performs well on average, and has some nice advantages such as no mention-candidates dictionary and simplicity of generalization to out-of-domain tasks.

**Summary Of The Review:**

Solid paper about an original approach to entity linking that performs pretty well.

---

> ### Author Response · Authors · 2021-11-19
> **Author Response**
>
> We thank the reviewer for the positive review and for the constructive comments. The reviewer makes a great point about possibly improving over the current basic thresholding scheme. To address the reviewer’s point, we have automatically calibrated the threshold value based on the AIDA validation performance by considering every effective threshold obtained from a sorted list of probabilities of labeled mentions. The best threshold is 0.03146, compared to 0.05 we used previously. The validation F1 scores are
>
> - Static threshold (0.05): 87.48
> - Dynamic threshold (0.03146): 87.75
>
> And the GERBIL test scores are (we further improved our baseline EntQA performance from 60.21 to 60.46 by better optimization)
>
> - Static threshold (0.05): 60.46
> - Dynamic threshold (0.03146): 60.55
>
> So it improves marginally, but overall we find that EntQA is robust to choices of threshold in a reasonable range. We will include these results in the final version of the paper.
>
> #1. We agree that the sentence can be made more clear. As the reviewer implies, the problem is that the search is approximate. Consider running GENRE with beam size 1. In this case, it is clear that the model must predict a mention before predicting its entity. Even with a larger beam size, each beam element will need to predict a mention before predicting an entity. Unless the beam size is unbounded so that every labeled span is considered, there will be errors from the MD phase propagating to ED. We will make this discussion more explicit.
>
> #2. We tried the first token, the first sentence, and a continuous topic embedding (obtained by averaging all token embeddings in the document). We settled on the first-token version because it gave the best performance. For many of the GERBIL datasets, we obtain almost the same performance with or without the topic information. As it is somewhat dataset-specific, we plan to leave it as an option in our model for the user to decide. The following shows the GERBIL performance without any topic information vs with the first token in the document:
>
> data|no topic|first token
> -----|-------|-------
> OKE16 |48.91|49.56
> AIDA|81.67|85.75
> Der|52.50|53.54
> R128|54.03|54.11
> OKE15|59.01|58.85
> MSNBC|72.23|70.99
> K50|64.20|67.22
> R500|40.32|41.65
> avg|59.11|60.21
>
> #3. We use one of the unused token types in the vocabulary and insert it in between two texts being concatenated before applying the encoder.

---

> > ### Comment · Reviewer_Uses · 2021-11-19
> > **Thank you**
> >
> > Thank you for the thorough answers and additional experiments.

---

### Official Review · Reviewer_SZGo · 2021-11-01

**Correctness:** 3
**Technical Novelty And Significance:** 3
**Empirical Novelty And Significance:** 3
**Recommendation:** 8
**Confidence:** 4

**Main Review:**

Strength:
* The novel entity linking paradigm proposed by this paper is insightful and the performance gain is significant.
* The proposed paradigm addresses a key issue in prior entity linking works. In EntQA, mention detection happens after entity detection, so that mention detection can utilize entity information.
* The proposed QA paradigm allows EntQA to take advantage of existing question answering datasets. Pretraining on open domain question answering datasets may improve the generalizability of EntQA. So not surprisingly, EntQA performs better on out-of-domain datasets.
* The proposed method has several practical advantages. It costs less GPU hours and memory space in comparison with prior SOTA according to the paper. However, providing the exact number will be much better.

Weakness:
* When discussing the difference between entity linking paradigms, this paper doesn’t compare with GENRE [1], although it is listed as a baseline. As the main contribution of this paper is the novel paradigm, I would like to see a section instead of a brief paragraph in introduction to discuss the theoretical and empirical difference between different paradigms. However, considering that Table 1 already shows the proposed paradigm performs better than other paradigms empirically, the theoretical analysis can be left for future work.
* The term ‘unknown entities’ is confusing, because it can also refer to entities not covered by KB. I suggest the authors clarify the definition. I think it is better to say ‘predict mentions when not knowing the corresponding entities’ than ‘predict mentions of unknown entities’. Besides, I wonder how mentions of entities not covered by KB will influence different paradigms, although it may be a minor issue.
* The effectiveness of EntQA is only shown by overall performance. It would be better to have experiments to compare the mention detection results of different methods.

[1] De Cao, N., Izacard, G., Riedel, S., & Petroni, F. (2020, September). Autoregressive Entity Retrieval. In International Conference on Learning Representations.


**Summary Of The Paper:**

This paper proposes a new paradigm for entity linking. The proposed method EntQA retrieves candidate entities first, and then uses candidate entities as queries to find mention spans.
EntQA addresses the issue of finding mentions without knowing their entities and offers many practical advantages. Experimental results show that EntQA can achieve good performance without mention-candidates dictionaries and large-scale task-specific weak supervision.


**Summary Of The Review:**

The insights and novel paradigm for entity linking proposed by this paper are attractive. Although the experiments can be more sufficient, the key claims are supported by the experimental results. I recommend accepting this paper.

---

> ### Author Response · Authors · 2021-11-19
> **Author Response**
>
> We thank the reviewer for the positive review, and for the great suggestion about the phrase “mentions of unknown entities”. We agree that this can indeed be confusing and will make the description more explicit as the reviewer recommends. We note that the draft already contains some details and discussions that the reviewer is asking for. We give the exact time and the number of GPUs in training in Section 3.1 (Model details). We discuss the pros and cons of EntQA vs GENRE in Section 4 (last paragraph). We isolate and analyze MD performance in Section 3.4 (last paragraph); we consider comparing the MD performance of other models to be out of scope. We will strengthen these sections and make them more clear as the reviewer recommends.

---

> > ### Comment · Reviewer_SZGo · 2021-11-20
> > **Thanks for the response**
> >
> > Sorry for missing some details and not making my points clear.
> >
> > For GPU hours and MD results, I think the comparison between EntQA and other methods is necessary because this paper claims that EntQA is more efficient in practice and addresses the dilemma of having to predict mentions before what they refer to. Reporting the results of EntQA solely is not enough to support the claims.
> >
> > For comparison between EntQA and GENRE, I think it's better to have a discussion of paradigms (section 4 only discusses practical details). Because GENRE is different from previous paradigms and do not have separate MD and ED stages.

---

> > > ### Author Response · Authors · 2021-11-20
> > > **Follow-up**
> > >
> > > Thank you for the clarifications. We agree that it will be more complete to analyze other models in addition to EntQA. As for training efficiency, our main point of comparison is GENRE which requires pretraining on 64 GPUs for 30 hours (Section 3.2.1).
> > >
> > > We will include more details in discussing EntQA and GENRE as paradigms. Even though GENRE does not separate MD and ED, it must predict a mention first before generating an entity for that mention.

---

### Official Review · Reviewer_97R7 · 2021-11-02

**Correctness:** 4
**Technical Novelty And Significance:** 3
**Empirical Novelty And Significance:** 2
**Recommendation:** 8
**Confidence:** 3

**Main Review:**

This paper addresses a problem of entity linking. The authors propose a reformulation of the problem as an inverted open-domain Question-Answers (QA) The authors used dense retriever in the first phase of their algorithm (retrieving top K entities) and then in the second phase authors trained reader mouse for ranking and extracting entity mentions from the output of the retriever.
The main advantages of the proposed method (besides outperforming previous strategies) is the data efficiency gained by eliminating dependency on hardcoded mention-candidate dictionary. Instead it finds entities from text passages and then finds lined mentions in test passages.
Authors work comes with strong backing from trying they model on GERBIL benchmark platform (trying it on various datasets) I especially appreciated the authors explaining the reason why their method did not do as well on OKE15/16 datasets compared to how it performed on other datasets.
Regarding the reproducibility of the results, the authors did not share their source code so it might take some effort to reproduce what they have done. On the other hand, they provided very detailed description of their training process including very helpful Model details section with lot's of relevant details parameters used with training and inference.
Authors provided very insightful discussion of their results. In addition to what's been provided I would be interested to learn how quick is the inference and how the time to run inference is impacted by the choice of K.

**Summary Of The Paper:**

This paper improves on the stat-of-the-art in entity linking.  Entity linking involves connecting entities to their mentions in test passages.
Current solutions rely on finding mentions first before running entity disambiguation stage which is difficult because mentions are being searched for unknown entities in the first stage.
The authors address this challenge by rearranging the standard QA pipeline by learning questions first (finding entities) and searching for answers (mentions) at a later stage.
This approach appears to be well-researched. The authors tested their model on GERBIL benchmarking platform and achived very positive results.


**Summary Of The Review:**

Overall, I believe this paper is a worthy addition to this year's conference. It tackles an important problem and provides a better performing, simpler and more intuitive solution compare to previous work.

---

> ### Author Response · Authors · 2021-11-19
> **Author Response**
>
> We thank the reviewer for the positive review. We will release the source code and make our results easily reproducible upon publication. The reviewer makes a great point about adding a discussion on the inference runtime. To address the reviewer’s point, we have compared EntQA with GENRE in inference time on the AIDA validation set (each using 1 GPU on the same machine). GENRE takes 1 hour and 10 minutes, excluding 31 minutes to first build a prefix tree. For EntQA, the runtime is linear in K, and we have
>
> - K=100: 1 hour and 36 minutes, validation F1 87.5
> - K=50: 49 minutes, validation F1 87.4
> - K=20: 20 minutes, validation F1 87.0
>
> Interestingly, we can obtain a significant speedup at a minor cost in performance by decreasing the number of candidate entities, which may be another useful feature of the model in controlling the speed-performance tradeoff. We will include this analysis in the final version.

---

### Public Comment · ~Nicola_De_Cao1 · 2021-11-10
**Two statements need corrections**

I wanted to point that the following two statements (from Section 3.2.1) are not completely right:
> While not mentioned in the paper or on the GitHub repository, GENRE (Cao et al., 2021) also uses the dictionary from Kolitsas et al. (2018) in their prefix tree to constrain the beam search (personal communication with one of the authors of the paper).

In GENRE (De Cao et al., 2021), specifically in Section 4.1 authors report that "we reproduce the setting of Kolitsas et al. (2018)". So they mention that they use the same setting (eg, dictionary, train/valid/test splits etc.) of Kolitsas et al. (2018).

> Constructing such a dictionary typically assumes the existence of a large KB-specific labeled corpus (e.g., internal links in Wikipedia). EntQA is thus more broadly applicable to KBs without such resources (e.g., for small domain-specific KBs).

This is not true as GENRE can also make predictions without mention-entity candidates. This is shown in the Ablation study and Appedix B.1.

---

> ### Author Response · Authors · 2021-11-19
> **GENRE without a dictionary**
>
> Thank you for your comment. It is certainly true that GENRE can be run without a mention-candidates dictionary, but the performance suffers severely. We ran GENRE without a dictionary and obtained the following results on GERBIL
>
> model|AIDA-B | MSNBC | Der | Kore50 | R128 | R500 | OKE15 | OKE16 | avg
> ---|---------|---------|---------|---------|---------|---------|---------|---------|---------
> w/ dictionary|  83.7 |73.7 |54.1 |60.7 |46.7 |40.3 |56.1 |50.0 |58.2
> w/o dictionary|  62.2 | 48.0 | 47.5 | 42.1 | 33.9 | 33.9 | 46.9 |45.1 | 45.0
>
> In comparison, EntQA obtains 60.5 avg without using any dictionary.

---

> > ### Public Comment · ~Nicola_De_Cao1 · 2022-02-05
> > **Wrong comparison**
> >
> > This is an unfair comparison: you are saying that when GENRE doesn't have a mention-candidate pairs it is as it would not have the list of all entities is the the KB. When GENRE doesn't have candidates you don't have to run it without constrained generation. You still run constrained generation where the candidates is the list of all the millions of possible KB entities. Otherwise even your model cannot classify without knowing the set of labels.
> >
> > Therefore even without resources such Wikipedia and even in unsupervised domains you can still run GENRE without candidates. What you wrote in the paper about the difficult applicability of GENRE is not true.

---

### Decision · Program_Chairs · 2022-01-20

**Decision:**

Accept (Spotlight)

**Comment:**

This paper casts entity linking in a retrieve-then-read framework by first retrieving entity candidates and then finding their mentions via reading comprehension. All reviewers agree that the proposed approach is novel, well-motivated, and simple yet performant. The authors have done a good job of addressing all the concerns raised, and the reviewers are unanimous in their recommendation for accepting the paper. I hope the authors will also incorporate the feedback and their responses in the final version.